# Prevalence of anemia in predialysis chronic kidney disease: Is the study center a significant factor?

Selma Alagoz[1], Mevlut Tamer Dincer[2], Necmi Eren[3], Alev Bakir[4], Meltem Pekpak[2], Sinan Trabulus[2], Nurhan Seyahi[2]*

1 Department of Nephrology, Bagcilar Training and Research Hospital, University of Health Sciences, Istanbul, Turkey, 2 Department of Nephrology, Cerrahpasa Medical Faculty, Istanbul University-Cerrahpasa, Istanbul, Turkey, 3 Department of Nephrology, Kocaeli University Medical Faculty, Kocaeli, Turkey, 4 Department of Biostatistics and Medical Informatics, Faculty of Medicine, Halic University, Istanbul, Turkey

* nseyahi@yahoo.com

**Data Availability Statement:** All relevant data are within the manuscript and its Supporting Information files.

## Abstract

### Objectives

Anemia is highly prevalent in chronic kidney disease patients; however, its identification and management have been reported to be suboptimal. In this study we aimed to describe the prevalence, severity, risk factors, and treatment of anemia in different nephrology centers, among chronic kidney disease patients who were not given renal replacement therapy.

### Materials and methods

We performed a multicenter cross-sectional study in three different nephrology clinics. Adult (>18 years of age) chronic kidney disease patients with an estimated glomerular filtration rate (eGFR) below 60 ml/min, and who were not started dialysis were recruited. Demographic, clinical and laboratory data regarding anemia and its management were collected using a standard data form. Anemia was defined as a hemoglobin level below 12g/dL and severe anemia as a hemoglobin level below 10g/dl.

### Results

A total of 1066 patients were enrolled in the study. Anemia and severe anemia were present in 55.9% and 14.9% of the patients, respectively. The mean hemoglobin level for the whole cohort was 11.8±1.8 g/dL. Univariate analyses revealed that the mean hemoglobin level was significantly different among the centers. Moreover, the frequency of the presence of anemia stratified by severity was also significantly different among the centers. According to binary logistic regression analysis, gender, levels of eGFR and iron, ferritin $\geq$ 100 ng/mL, and the nephrology center were independent determinants of severe anemia.

### Conclusions

We found a high prevalence of anemia among chronic kidney disease patients who were not on renal replacement therapy. Each center should determine the treatment strategy

**Funding:** The authors received no specific funding for this work.

**Competing interests:** The authors have declared that no competing interests exist.

according to the patient's characteristics. According to our results, the center-specific management of anemia seems to be important.

## Introduction

Anemia is a highly prevalent and modifiable risk factor for many adverse events in patients with chronic kidney disease (CKD) [1]. Anemia also contributes to the progression of CKD [2]. The greatest declines in the hematocrit level are observed in the early stages of kidney disease, with the reductions getting smaller in moderate to advanced renal failure. Thus, early detection and monitoring of anemia are required in CKD patients [3].

A significant increase in the prevalence of anemia develops as the creatinine clearance falls below 70 mL/min in males or below 50 mL/min in females [2]. The correction of anemia has been shown to improve cardiac and cognitive functions, quality of life, physical activity, shorten the hospitalization period and decrease mortality [4–8]. Despite these benefits, identification, and management of anemia among patients with CKD has been reported to be suboptimal. Anemia in CKD patients on dialysis has been extensively studied. However, in CKD patients who are not yet on hemodialysis, there is a paucity of large-scale studies [1,2]. Moreover, optimal management of anemia in predialysis patients remains uncertain [9]. According to a large-scale randomized control trial, performed in predialysis CKD patients, hemoglobin (Hgb) normalization (Hgb≥13 g/dL) was associated with increased mortality [10]. However, a recent meta-analysis favors a higher Hgb target in predialysis patients [11]. Additionally, predialysis management of anemia with erythropoiesis-stimulating agents (ESA) was found to be associated with reduced all cause and cardiovascular mortality in patients attaining a Hgb level of >9 g/dL.

According to a recent large-scale multicenter multinational study, there is a striking difference between different countries regarding the frequency of predialysis anemia [12]. However, to the best of our knowledge, center-based differences were not extensively studied previously. We performed a study to describe the prevalence, severity, risk factors, and treatment of anemia among CKD patients who were not given renal replacement therapy in different nephrology centers. We also aimed to analyze the center-based differences regarding those parameters.

## Patients and methods

The study was approved by the Clinical Research Ethics Committee of Cerrahpasa Medical Faculty (approval number: 117945/2018). All participants gave written informed consent. We performed a multicenter cross-sectional study in three different nephrology clinics located in the same geographical region (Marmara) of Turkey. Center A and B (Old Town) are located in Istanbul and Center C is located in Kocaeli. The number of inpatient bed for nephrology were 10 in Center A, 15 in Center B and 29 in Center C. The total number of inpatient beds for all departments were 500 in Center A, 1350 in Center B and 730 in Center C. The number of patients who applied to the outpatient nephrology clinic in a month were approximately 1300 in Center A, 1350 in Center B and 1130 in Center C. A total of 1066 CKD patients who were >18 years of age, had an estimated glomerular filtration rate (eGFR) below 60 ml/min, were not started dialysis and were under regular follow-up at the outpatient clinics were included in this study. The study was conducted between February 2018 and August 2018.

All consecutive patients who met the inclusion criteria of the study during the enrollment period were included and data were collected using a standard data form. Baseline data included sex, age, causes of kidney disease, presence of menopause, diabetes mellitus, primary hematologic disease, and malignancy. Blood samples were analyzed at the respective laboratories of the participating centers. Laboratory data, including complete blood cell counts, serum creatinine, C-reactive protein (CRP), vitamin B12, folate, ferritin, iron, total iron-binding capacity (TIBC), transferrin saturation ratio, and intact parathyroid hormone (iPTH) were collected from the medical records. Current use of iron supplements, ESA, folate, and vitamin B12 supplements were also recorded. Anemia was defined as a Hgb level below 12 g/dL and severe anemia was defined as a Hgb level below 10 g/dl [6]. Glomerular filtration rate was estimated using the abbreviated version of the Modification of Diet in Renal Disease (MDRD) formula [13].

Our study had the following potential bias inheriting to the cross-sectional study design. First, there is a possibility of information bias, since we collected the data on drug use not just through the medical records but with patient interviews. Second, there is a possibility of selection bias, because we examined patients during a time period of seven months. We probably missed a proportion of the patients who attended to outpatient clinics less frequently, such as, once a year. The study was conducted in accordance with the principles of the 1975 Declaration of Helsinki (as revised in 1983).

## Statistical analysis

The characteristics of the patients were described using descriptive statistics; categorical data were stated as counts and proportions, and continuous data as mean standard deviation (SD), median and minimum-maximum values. The statistical differences between the groups were calculated using the chi-square test for nominal variables. The distribution normality of quantitative variables was calculated with the Shapiro-Wilk test. We compared the groups using one-way ANOVA for normally distributed variables, or otherwise using the Kruskal-Wallis test. Post-hoc multiple comparison analysis was performed with significant values adjusted by the Bonferroni correction. Binary logistic regression analysis was used to predict the association of covariate variables with severe anemia. The presence of malignancy and primary hematological disease (PHD) have been associated with anemia [14]. Therefore, we excluded the patients with malignancy and PHD from the logistic regression analysis, and accordingly, this analysis was carried out in 971 subjects. We constructed a multivariate model using the variables selected according to the P value ($<$0.05) of univariate analysis (S1 Table). The following variables were selected: gender, nephrology center, presence of diabetes, ferritin $\geq$ 100 ng/mL, levels of eGFR, CRP, iPTH, iron, and TIBC. The stage of CKD and creatinine is reflected by eGFR, therefore, this parameter was not included in the binary logistic regression analysis. Additionally, since ESA and iron were used because of anemia, these parameters were also not included in the binary logistic regression analysis. Statistical analysis was performed using the IBM SPSS v.24 for Windows software and was reported with 95% confidence intervals (CI). Values of p$<$0.05 were considered significant.

## Results

### General characteristics of the study population

The general characteristics of the patients are shown in Table 1. Study subjects were generally old patients (median age: 68.0, range: 18–97) with a nearly equal gender distribution. Most of the females were in menopause, and diabetes was present in nearly half of the patients. Primary etiologies of CKD were diabetes mellitus (47.1%) and hypertension (27.8%), followed by glomerulonephritis (3.3%), and polycystic kidney disease (2.1%). Etiology could not be

**Table 1. General characteristics of the patients according to the center.**

| | All Centers n = 1066 | Center A n = 447 | Center B n = 398 | Center C n = 221 | p-value |
|---|---|---|---|---|---|
| Age (years) | 65.5±13.7 | 67.3±12.3 | 65.9±13.4 | 61.0±15.8 | <0.001 |
| Gender (male), n (%) | 517 (48.5) | 219 (49.0) | 167 (42.0) | 131 (59.3) | <0.001 |
| DM, n (%) | 502 (47.1) | 237 (53.0) | 182 (45.7) | 83 (37.6) | 0.001 |
| Malignancy, n (%) | 77 (7.2) | 25 (5.6) | 30 (7.5) | 22 (10.0) | 0.117 |
| Menopause, n (%) | 486 (88.5) | 207 (90.8) | 209 (90.5) | 70 (77.8) | 0.002 |
| PHD, n (%) | 24 (2.3) | 5 (1.1) | 11 (2.8) | 8 (3.6) | 0.084 |
| Hgb (g/dL) | 11.8±1.8 | 11.6±1.9 | 12.0±1.7 | 11.7±1.8 | <0.001 |
| Htc (%) | 36.0±5.4 | 36.3±5.7 | 36.3±5.0 | 35.1±5.4 | 0.024 |
| MCV (fL) | 86.1±6.0 | 87.0±6.1 | 85.6±5.6 | 84.9±6.3 | <0.001 |
| Hgb (g/dL) | | | | | |
| Hgb<10, n (%) | 159 (14.9) | 86 (19.2) | 38 (9.5) | 35 (15.8) | |
| 10≤Hgb<12, n (%) | 437 (41.0) | 184 (41.2) | 160 (40.2) | 93 (42.1) | <0.001 |
| Hgb≥12, n (%) | 470 (44.1) | 177 (39.6) | 200 (50.3) | 93 (42.1) | |
| Creatinine (mg/dL) | 2.2±1.2 | 2.3±1.3 | 1.9±1.1 | 2.3±1.4 | <0.001 |
| eGFR (mL/min/1.73m$^2$) | 35.8±14.2 | 33.4±13.7 | 38.6±13.6 | 35.7±15.2 | <0.001 |
| eGFR | | | | | |
| Stage 3 (30–59) | 694 (65.1) | 262 (58.6) | 294 (73.9) | 138 (62.4) | <0.001 |
| Stage 4 (15–29) | 279 (26.2) | 138 (30.9) | 82 (20.6) | 59 (26.7) | |
| Stage 5 (<15) | 93 (8.7) | 47 (10.5) | 22 (5.5) | 24 (10.9) | |
| CRP (mg/L) | 12.9±23.3 | 14.6±25.6 | 8.6±13.7 | 17.3±29.9 | <0.001 |
| iPTH (pg/mL) | 137.8±151.6 | 144.1±137.4 | 115.2±108.2 | 175.2±243.4 | <0.001 |
| Iron (μg/dL) | 64.8±30.2 | 62.9±29.5 | 67.8±30.4 | 62.1±31.5 | 0.012 |
| TIBC (μg/dL) | 282.8±80.1 | 254.1±82.8 | 310.7±63.4 | 296.0±83.3 | <0.001 |
| TSAT (%) | 25.8±20.5 | 29.9±27.1 | 22.6±11.2 | 21.8±12.4 | <0.001 |
| TSAT<%20, n (%) | 423 (43.6) | 175 (39.4) | 180 (45.8) | 68 (50.7) | 0.035 |
| Ferritin (ng/mL) | 148.3±225.2 | 122.6±141.7 | 164.7±244.9 | 175.6±321.7 | 0.001 |
| Ferritin<100 (ng/mL), n (%) | 585 (57.5) | 266 (59.9) | 209 (53.3) | 110 (60.8) | 0.097 |
| Vit B12 (pg/mL) | 369.5±280.6 | 332.5±292.2 | 435.2±282.3 | 312.6±194.9 | <0.001 |
| Folate (ng/mL) | 8.6±5.4 | 8.2±4.5 | 8.7±4.8 | 9.6±8.7 | 0.671 |
| Vit B12 use, n (%) | 191 (17.9) | 85 (19.0) | 54 (13.6) | 52 (23.5) | 0.006 |
| Folate use, n (%) | 75 (7.0) | 34 (7.6) | 28 (7.0) | 13 (5.9) | 0.715 |
| Iron use, n (%) | 278 (26.1) | 145 (32.4) | 75 (18.8) | 58 (26.2) | <0.001 |
| ESA use, n (%) | 117 (11.0) | 79 (17.7) | 9 (2.3) | 29 (13.1) | <0.001 |

Values are presented as mean±standard deviation for the continuous variables and frequency (percentage) for the categorical variables.

CRP: C-reactive protein, DM: diabetes mellitus, eGFR: estimated glomerular filtration rate, ESA: erythropoiesis- stimulating agent, Hgb: hemoglobin, Htc: hematocrit, iPTH: intact parathyroid hormone, MCV: mean corpuscular volume, PHD: primary hematological disease, TIBC: total iron binding capacity, TSAT: transferrin saturation ratio, Vit B12: vitamin B12.

detected in 11.0% of the patients. Various causes of CKD, such as nephrolithiasis, vasculitis, uric acid nephropathy, vesicoureteral reflux, pyelonephritis, Alport syndrome, renal tumor, and amyloidosis were reported in the remaining 8.7% of the patients.

Most of the patients had Stage 3 or 4 CKD. Patient characteristics stratified by the study center are shown in Table 1. There were statistically significant differences between the study centers regarding study parameters. However, the frequency of malignancy, primary hemato-logical disease (PHD), ferritin below 100 ng/mL, folate use, and folate levels were similar between the study centers.

**Table 2. Baseline characteristics of the patients according to hemoglobin levels.**

| | Hg<10 g/dL (n = 159) | 10≤Hg<12 g/dL (n = 437) | Hg≥12 g/dL (n = 470) | p-value |
|---|---|---|---|---|
| Age (years) | 66.1±14.2 | 65.6±13.1 | 65.2±14.0 | 0.676 |
| Gender (male), n (%) | 63 (39.6) | 178 (40.7) | 276 (58.7) | <0.001 |
| DM, n (%) | 85 (53.5) | 223 (51.0) | 194 (41.3) | 0.003 |
| Malignancy, n (%) | 21 (13.2) | 27 (6.2) | 29 (6.2) | 0.007 |
| Menopause, n (%) | 84 (87.5) | 232 (89.6) | 170 (87.6) | 0.766 |
| PHD, n (%) | 10 (6.3) | 8 (1.8) | 6 (1.3) | 0.005 |
| Hgb (g/dL) | 9.2±0.7 | 11.0±0.6 | 13.4±1.2 | <0.001 |
| Htc (%) | 28.4±2.6 | 33.8±2.1 | 40.7±3.7 | <0.001 |
| MCV (fL) | 85.9±7.6 | 85.4±6.1 | 86.7±5.3 | 0.001 |
| Creatinine (mg/dL) | 3.0±1.6 | 2.2±1.3 | 1.8±0.8 | <0.001 |
| eGFR (mL/min/1.73m$^2$) | 25.6±13.8 | 34.3±13.9 | 40.8±12.2 | <0.001 |
| eGFR | | | | |
| Stage 3 (30–59) | 48 (30.2) | 264 (60.4) | 382 (81.3) | <0.001 |
| Stage 4 (15–29) | 74 (46.5) | 132 (30.2) | 73 (15.5) | |
| Stage 5 (<15) | 37 (23.3) | 41 (9.4) | 15 (3.2) | |
| CRP (mg/L) | 22.5±31.7 | 13.2±25.3 | 9.3±15.5 | <0.001 |
| iPTH (pg/mL) | 196.2±157.2 | 140.2±136.5 | 115.8±158.0 | <0.001 |
| Iron (μg/dL) | 57.8±35.1 | 60.9±29.4 | 71.0±27.9 | <0.001 |
| TIBC (μg/dL) | 246.8±95.6 | 279.0±71.1 | 299.4±77.8 | <0.001 |
| TSAT (%) | 0.29±0.29 | 0.24±0.17 | 0.26±0.19 | <0.001 |
| TSAT<20%, n (%) | 66 (44.3) | 196 (48.6) | 161 (38.4) | <0.013 |
| Ferritin (ng/mL) | 282.3±378.2 | 147.4±224.8 | 103.3±107.2 | 0.001 |
| Ferritin<100 (ng/ml), n (%) | 53 (34.9) | 248 (59.0) | 284 (63.8) | <0.001 |
| Vit B12 (pg/mL) | 410.8±307.4 | 385.3±297.0 | 339.5±250.4 | 0.002 |
| Folate (ng/mL) | 8.7±6.4 | 9.0±6.1 | 8.2±4.3 | 0.329 |
| Vit B12 use, n (%) | 27 (17.0) | 90 (20.6) | 74 (15.7) | 0.155 |
| Folate use, n (%) | 11 (6.9) | 38 (8.7) | 26 (5.5) | 0.176 |
| Iron use, n (%) | 67 (42.1) | 145 (33.2) | 66 (14.0) | <0.001 |
| ESA use, n (%) | 56 (35.2) | 45 (10.3) | 16 (3.4) | <0.001 |

Values are presented as mean±standard deviation for the continuous variables and frequency (percentage) for the categorical variables.

CRP: C-reactive protein, DM: diabetes mellitus, eGFR: estimated glomerular filtration rate, ESA: erythropoiesis- stimulating agent, Hgb: hemoglobin, Htc: hematocrit, iPTH: intact parathyroid hormone, MCV: mean corpuscular volume, PHD: primary hematological disease, TIBC: total iron binding capacity, TSAT: transferrin saturation ratio, Vit B12: vitamin B12.

## Anemia parameters and the use of erythropoiesis-stimulating agents and iron

Anemia and severe anemia were present in 55.9% and 14.9% of the patients, respectively. The mean Hgb level for the whole cohort was found 11.8±1.8 g/dL. The baseline characteristics of the patients according to Hgb levels are shown in Table 2.

According to multi-group comparisons, anemia was associated with the female gender, the presence of diabetes, malignancy and PHD. We also showed the well-known association between decreasing eGFR and the presence of anemia. Fig 1 demonstrates the negative association between the prevalence of anemia and eGFR, indicating that the percentage of the patients with anemia increases while kidney function decreases. The percentage of the patients with Hgb greater than 12 g/dL was significantly higher in Stage 3 than Stage 4 and 5 CKD patients (55.1% vs 26.2% and 16.1%, respectively, p<0.001). Conversely, the percentage of the patients

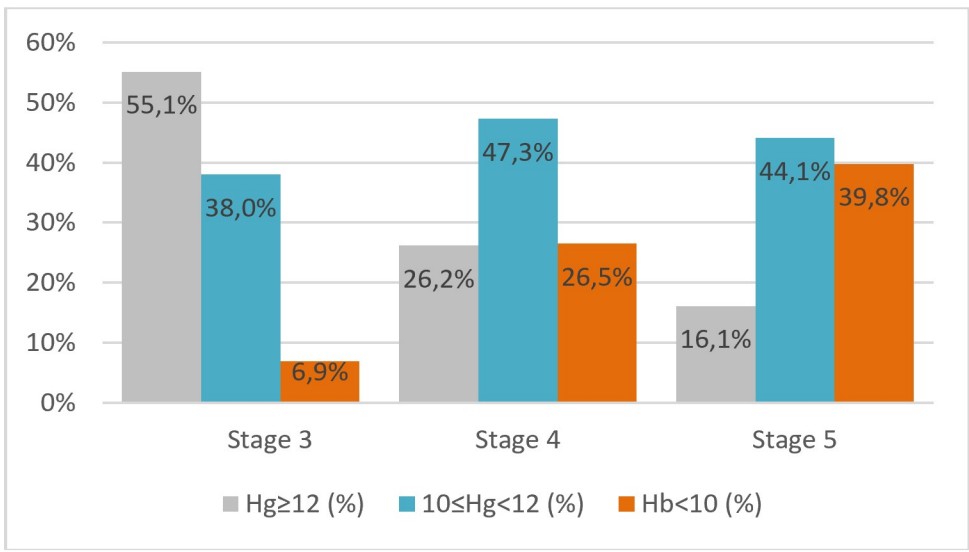

**Fig 1. Prevalence of anemia based on staging of CKD.**

with Hgb<10 g/dL was significantly lower in Stage 3 than Stage 4 and 5 CKD (6.9%vs 26.5% and 39.8%, respectively, p<0.001).

Regarding iron-related parameters, there was a trend toward higher iron use in patients with anemia. In line with this finding, ferritin levels tended to be higher in patients with anemia, possibly at least partially reflecting the effect of iron treatment. Distribution of the patients with TSAT below 20% grouped according to Hgb levels was also not homogenous.

The use of ESA was also associated with the presence of anemia. We want to point out that ESA was used only by 56 of the 159 patients with severe anemia. Malignancy or PHD was present in 26 of these 159 patients. Additionally, iron deficiency defined as a ferritin level <100 ng/ml and a TSAT level <%20 were present in 28 patients. Therefore, the remaining 53 patients were candidates for ESA treatment.

## Binary logistic regression

We used binary logistic regression analysis to examine the independent variables associated with severe renal anemia. According to multivariate forward-stepwise binary logistic regression analysis, gender (p = 0.027; OR: 1.637; 95% CI: 1.058–2.533), eGFR (p<0.001; OR: 0.951; 95% CI: 0.935–0.967), ferritin $\geq$ 100 ng/ml (p = 0.001; OR: 2.144; 95% CI: 1.368–3.362), iron (p = 0.041; OR: 0.991; 95% CI: 0.983–0.999), and the center (p = 0.005) were independent determinants of severe anemia (Table 3). It should be noted that the CRP level was also associated with severe anemia in logistic regression, with a borderline statistical significance (p = 0.064; OR: 1.008; 95% CI: 1.000–1.016).

## Discussion

In this multicenter, cross-sectional study, we evaluated the prevalence, severity, risk factors and treatment of anemia in CKD patients who were not given renal replacement therapy. Anemia and severe anemia were present in 55.9% and 14.9% of the patients, respectively. These data are consistent with previous studies. According to McClellan et al., in CKD patients with an eGFR level below 60 ml/min, the prevalence of anemia and severe anemia was 47.7% and 8.9%, respectively [2]. In another multicenter study, the prevalence of anemia and severe

**Table 3. Covariates associated with severe anemia.**

| | | Estimate | SE | p | OR | 95% CI for OR |
|---|---|---|---|---|---|---|
| Center | | | | 0,005 | | |
| | A | Reference | | | | |
| | B | -0.834 | 0.262 | 0.001 | 0.434 | 0.260–0.725 |
| | C | 0.020 | 0.312 | 0.950 | 1.020 | 0.553–1.880 |
| Gender, Male | | Reference | | | | |
| Female | | 0.493 | 0.223 | 0.027 | 1.637 | 1.058–2.533 |
| Ferritin < 100 | | Reference | | | | |
| Ferritin ≥ 100 | | 0,763 | 0.229 | 0.001 | 2.144 | 1.368–3.362 |
| CRP | | 0,008 | 0,004 | 0,064 | 1,008 | 1,000–1,016 |
| Iron | | -0.009 | 0.004 | 0.041 | 0.991 | 0.983–0.999 |
| e-GFR | | -0,050 | 0.009 | 0.000 | 0.951 | 0.935–0.967 |
| Constant | | 0.142 | 0.365 | 0.697 | 1.153 | |

CI: confidence interval, eGFR: estimated glomerular filtration rate, OR: odds ratio, SE: standard error

anemia in CKD patients with an eGFR level below 60 ml/min were found to be 38% and 7.5% respectively [12]. In line with previous studies, we observed an association between anemia and decreasing kidney function [2,15].

Additionally, we showed that the female gender, eGFR, serum levels of iron, and ferritin were independent risk factors for severe anemia. Furthermore, the center was an additional independent risk factor for severe anemia. The presence of diabetes mellitus was found to be a risk factor for severe anemia in univariate analysis, but not in multivariate analysis. Previous studies showed that female gender, history of diabetes mellitus, CKD stage, serum transferrin saturation, serum levels of ferritin and iPTH and angiotensin-converting enzyme inhibitors (ACEI) or angiotensin receptor blocking agent (ARB) were risk factors for severe anemia in CKD patients [15–19].

We have found different risk profiles of anemia for different medical centers. We want to point out that the specific nephrology center was found as an independent determinant of anemia according to our logistic regression model. Our study is one of the few studies investigating the effects of the medical center on anemia in CKD. Country-specific differences were previously reported [12]. However, to the best of our knowledge, center effect was not investigated for medical centers located in a single country. We think that the center effect on severe anemia may be due to the socioeconomic differences, along with nutritional characteristics, environmental factors, drug intake (such as ACEI or ARB), and self-care characteristics of the patients. Low socioeconomic background, different nutritional intakes, genetic, and environmental factors have been previously identified as risk factors for anemia [20–23]. These data from the literature suggest that interventions and iron intake guidelines should be tailored to regional, nutritional, and socioeconomic subgroups.

We also revealed that a substantial proportion of the potential candidates for ESA treatment were not using ESA. We suggest two potential explanations. First, our cross-sectional study might not capture forthcoming treatments. Second, "therapeutic inertia" might have a role. The therapeutic inertia is a well described concept in patients with CKD and this phenomenon might also be in charge in our cases [24].

There were several limitations to our study. First, a cross-sectional design is unable to capture the treatment effect as efficient as a longitudinal study. Another limitation was the use of three different laboratories and the lack of standardization between laboratories. Finally, we

did not assess the rates of blood transfusions and ACEI and ARB treatment. Additionally, generalizability of our results regarding anemia prevalence to a larger scale might be limited because of center-based differences.

In conclusion, we found a large prevalence of anemia among CKD patients who were not given RRT, and the burden of patients who require treatment with erythropoietin is considerably large. We found that some of these patients did not receive ESA treatment. Thus, there is a need to improve the timing of anemia intervention and the quality of care for these patients. Clinicians should be aware of this risk, identify and work up the anemic patients, and implement appropriate therapy. According to our results, center specific management of anemia seems to be important.

## Supporting information

**S1 Table. Baseline characteristics of the patients grouped according to the presence of severe anemia.**
(DOCX)

## Author Contributions

**Conceptualization:** Nurhan Seyahi.

**Data curation:** Selma Alagoz, Mevlut Tamer Dincer, Necmi Eren.

**Formal analysis:** Selma Alagoz, Alev Bakir.

**Investigation:** Selma Alagoz, Mevlut Tamer Dincer, Necmi Eren, Nurhan Seyahi.

**Supervision:** Nurhan Seyahi.

**Validation:** Meltem Pekpak.

**Writing – original draft:** Selma Alagoz.

**Writing – review & editing:** Selma Alagoz, Sinan Trabulus, Nurhan Seyahi.

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
