## [Decision Letter · Decision Letter 0]

24 Jan 2020

PONE-D-19-32342

Prevalence of anemia in predialysis chronic kidney disease: is the study center significant?

PLOS ONE

Dear Dr Seyahi,

Thank you for submitting your manuscript to PLOS ONE. After careful consideration, we feel that it has merit but does not fully meet PLOS ONE’s publication criteria as it currently stands. Therefore, we invite you to submit a revised version of the manuscript that addresses the points raised during the review process.

 ACADEMIC EDITOR:  

There are conflicts between the reviews, I decided to ask you for  major revisions.  In my opinion the article is interest but there are many papers already published on this topic. So authors should strengthen the manuscript providing more significant references, in order to reinforce the methodology part. Also provide references for "Anemia definition". The population among different centers is quite heterogeneous, authors should address this in the statistical analysis. Please answer to the criticisms moved by reviewers. 

We would appreciate receiving your revised manuscript by feb 14th. To enhance the reproducibility of your results, we recommend that if applicable you deposit your laboratory protocols in protocols.io, where a protocol can be assigned its own identifier (DOI) such that it can be cited independently in the future. For instructions see: http://journals.plos.org/plosone/s/submission-guidelines#loc-laboratory-protocols

We look forward to receiving your revised manuscript.

Kind regards,

Martina Crivellari

Academic Editor

PLOS ONE

Journal Requirements:

a) Did participants provide their written or verbal informed consent to participate in this study?

Reviewers' comments:

Reviewer's Responses to Questions

**Comments to the Author**

1. Is the manuscript technically sound, and do the data support the conclusions?

Reviewer #1: Yes

Reviewer #2: Yes

Reviewer #3: Yes

Reviewer #4: Yes

2. Has the statistical analysis been performed appropriately and rigorously? 

Reviewer #1: Yes

Reviewer #2: Yes

Reviewer #3: Yes

Reviewer #4: No

3. Have the authors made all data underlying the findings in their manuscript fully available?

Reviewer #1: Yes

Reviewer #2: Yes

Reviewer #3: Yes

Reviewer #4: Yes

4. Is the manuscript presented in an intelligible fashion and written in standard English?

Reviewer #1: Yes

Reviewer #2: Yes

Reviewer #3: Yes

Reviewer #4: No

5. Review Comments to the Author

Reviewer #1: This work is very good to look for the anemia in CKD Patients in different stages improving of the anemia for sure will have a benefit to the health service. By improving the anemia many chronic illness will be avoided.

Reviewer #2: Article is interested but need to clarify few points;

1. [ABSTRACT]; Title and even apart of objective indicated that centre and or centres based treatment / services will affect the treatment, but abstract results didn’t indicated anything like that. I will suggest to incorporate related findings, if possible. Otherwise, rephrase title or objective part………………………………………………………………………………………………………………..

2. [INTRODUCTION]; Introduction last sentence is different than title and even with abstract objective…………………………………………………………………………………………………………………………...

3. This section is not sufficient to provide enough background of the objectives, it is suggested to elaborate……………………………………………………………………………………………………………………….

4. What was the inclusion criteria for this current project………………………………………………..…….

5. To determine the incidence by getting this 6-7 months data from THREE centres are sufficient..? Shall we generalized this Incidence at country level…?........................................

6. Baseline data included: sex, age, causes of kidney disease, presence of menopause, diabetes mellitus, primary hematologic disease, and malignancy… Why only these factors, how about hypertension?.............................................................................................................................

7. Provide reference for the statement of “Anemia was defined by hemoglobin (Hgb) level below 12g/dL and severe anemia was defined by Hgb level below 10g/dl”………………………….

8. MDRD formula is practicing in the centre or any other reason to select this formula……………

9. Table 1; It would be looking more better than existing , if author convert this into descriptive statistics, as three centres comparison might not be that much fruitful sound. But, Still authors like to keep as it is then they have to provide all three centres background. I mean all three centres are associated with tertiary level referral hospitals and how much bed size and CKD population are there. Further, where these centres are located, and the general population….?.......................................................................................................................

10. It is suggested to avoid abbreviations in the heading/sub-heading, like “ Anemia Parameters and the use of ESA and Iron”………………………………………………………………………………..

11. Table 2; P supposed to be p-value…………………………………………………………………………

12. Conclusion should be more specific based on study findings………………………………..

Reviewer #3: This multicenter cross-sectional study is presented in an intelligible fashion, although it needs some minor English revision.

The authors state that the center is an independent risk factor for anemia in patients affected by predialysis chronic kidney disease. Anyway it is clear from the manuscript that the population among the centers is very dishomogeneus, with statistical significant differences in relevant characteristics such as gender distribution, eGFR, ferritin and iron blood level.

How the authors address this in their statistical analysis?

The following are minor comments, based on STROBE checklist:

9) BIAS, in the methods section, potential sources of bias are not addressed

10) Explain how you arrived at this study size

12.a) you should be more precise to describe the logistic regression: you decide to exclude patients with malignacy and PHD form the analysis. this should be stated in the methods section and the choice must be justified. the criteria used to select the variable for the multivariate analysis must be specified in the methods.

12.c) how do you handle missing data?

16) the presentation of the results of the binary logistic regression is not clear: OR (and not just p value should be reported in the text). You specified a reference level of ferritin < 100, but no other reference level for other continuous variable such as CRP e GFR and Iron are present.

18) your results shows that ESA is underused, this is interesting and should be cited in the discussion. Moreover, is there any significant difference among centers on this point?

20) the authors interpretate the impact of treatment center on anemia as due to the socio-economical differences in the population. it sounds strange if we think that the three center are located in the same area. Please clarify.

Reviewer #4: The authors have conducted an observational study to study prevalence of and management of anemia in non-dialysis dependent CKD patients. First the title of the study is very vague. Methodology section is very weak, there is no explanation of study design, study instrument, recruitment of patients, inclusion exclusion criteria, methodological flowchart etc. Statistical methods used in current study are very basic and gives very basic information. Authors could have performed more analysis especially for comparison of management in different centers with respect to patient outcomes. There are hundreds of articles on this topic online, i feel this authors literature review is bit weak. I would suggest them to read relevant articles and modify their paper accordingly.

6. PLOS authors have the option to publish the peer review history of their article (what does this mean?). If published, this will include your full peer review and any attached files.

Reviewer #1: Yes: faissal a. m. shaheen

Reviewer #2: Yes: AMER HAYAT KHAN

Reviewer #3: No

Reviewer #4: No

---

## [Author Response · Author response to Decision Letter 0]

15 Feb 2020

February 13, 2020

Martina Crivellari

Academic Editor

PLOS ONE

Dear Editor,

 We would like to thank you very much for your valuable review, which gave us the opportunity to improve our manuscript, titled “Prevalence of anemia in predialysis chronic kidney disease: is the study center a significant factor?”, MS number PONE-D-19-32342. In line with your suggestions, we made an extensive literature search to locate relevant and new references. We incorporated those literature to our revised manuscript and specifically provided references for “the definition of anemia”. Finally, we tried to present our statistical analysis more clearly and extensively in order to explain how we addressed the heterogenous population among different centers.

 We sincerely appreciate the useful and expert suggestions that helped us to prepare the revised manuscript. We thank you for your consideration and remain.

Yours truly,

Nurhan Seyahi, MD

Selma Alagoz, MD

PONE-D-19-32342

Prevalence of anemia in predialysis chronic kidney disease: is the study center a significant factor?

PLOS ONE

RESPONSE TO REVIEWERS

REVIEWER # 1 

This work is very good to look for the anemia in CKD Patients in different stages improving of the anemia for sure will have a benefit to the health service. By improving the anemia many chronic illness will be avoided.

Response : We thank the reviewer for those appreciative comments.

REVIEWER # 2

Article is interested but need to clarify few points;

1. [ABSTRACT]; Title and even apart of objective indicated that centre and or centres based treatment / services will affect the treatment, but abstract results didn’t indicated anything like that. I will suggest to incorporate related findings, if possible. Otherwise, rephrase title or objective part…….

Response: In the results section of the original abstract, we mentioned that nephrology center was an independent determinant of severe anemia (page 1, paragraph 3, line 6-7). Following the comments of the reviewer, we incorporated additional information to underline this subject in the revised manuscript (page 1, paragraph 3, line 3-5). 

2. [INTRODUCTION]; Introduction last sentence is different than title and even with abstract objective ……..

Response: Following the comments of the reviewer, we reedited the last part of the introduction to clarify the objective (page 2, paragraph 3, line 5-6). 

3. This section is not sufficient to provide enough background of the objectives, it is suggested to elaborate……

Response: In line with the comments of the reviewer, we provided additional information for the background of the objectives (page 2, paragraph 2, line 5-13; page 2, paragraph 3, line 1-3). We also included additional references to reinforce this section (reference no: 9,10,11).

4. What was the inclusion criteria for this current project…………

Response: As we wanted to reveal the frequency of anemia in a center, we aimed to recruit all adult consecutive stage G3-G5 CKD patients. 

5. To determine the incidence by getting this 6-7 months data from THREE centres are sufficient..? Shall we generalized this Incidence at country level…?....

Response: We agreed with the reviewer on his/her remark. In line with his/her comments, we did not generalize our incidence data to our country in the original manuscript. On the other hand, we recruited all patients to reveal the incidence of anemia in those centers.

Following the comments of the reviewer, we added information about the limitation of generalizability of our results (page 11, paragraph 4, line 5-6). 

6. Baseline data included: sex, age, causes of kidney disease, presence of menopause, diabetes mellitus, primary hematologic disease, and malignancy… Why only these factors, how about hypertension?..

Response: We focused to collect data on the factors related to anemia. Unfortunately, we did not collect data about the prevalence of hypertension. However, we mentioned about the frequency of hypertension as a case of CKD. 

7. Provide reference for the statement of “Anemia was defined by hemoglobin (Hgb) level below 12g/dL and severe anemia was defined by Hgb level below 10g/dl”… 

Response: We added relevant paper to the reference list in the revised manuscript (reference no: 6).

8. MDRD formula is practicing in the centre or any other reason to select this formula………

Response: We used the MDRD formula to estimate GFR, because, according to a previous study performed in our unit, the MDRD formula performed better than the other estimation equations, including CKD-EPI (Altiparmak MR, Ren Fail. 2013;35(8):1116-23. doi: 10.3109/0886022X.2013.817278).

9. Table 1; It would be looking more better than existing , if author convert this into descriptive statistics, as three centres comparison might not be that much fruitful sound. But, Still authors like to keep as it is then they have to provide all three centres background. I mean all three centres are associated with tertiary level referral hospitals and how much bed size and CKD population are there. Further, where these centres are located, and the general population….?..................

Response: Following the comments of the reviewer, we wanted to keep the Table 1 as it is, and then we provided background information for all tree centers in the relevant sections of the revised manuscript (page 3, paragraph 1, line 2-7).

10. It is suggested to avoid abbreviations in the heading/sub-heading, like “Anemia Parameters and the use of ESA and Iron”……

Response: We are sorry for this inconvenience; we corrected the abbreviations.

11. Table 2; P supposed to be p-value………………………………………

Response: The reviewer is correct in his/her assumption and we made the necessary change in the revised manuscript.

12. Conclusion should be more specific based on study findings………

Response: In our original manuscript, we made specific recommendations in the conclusion section regarding anemia management, such as improving the timing of anemia intervention and increasing the awareness of the clinicians.

We think that our results are not robust enough to make more specific recommendations. However, as suggested by the reviewer, there is a need for more specific recommendations regarding the management of CKD-related anemia in predialysis patients. We hope the results of our study will address this unmet need. 

REVIEWER # 3

This multicenter cross-sectional study is presented in an intelligible fashion, although it needs some minor English revision.

Response: In the revised manuscript, the whole text was reedited by a native English speaker.

The authors state that the center is an independent risk factor for anemia in patients affected by predialysis chronic kidney disease. Anyway, it is clear from the manuscript that the population among the centers is very dishomogeneous, with statistically significant differences in the relevant characteristics such as gender distribution, eGFR, ferritin and iron blood levels.

How the authors address this in their statistical analysis?

Response: The reviewer correctly stated that there are statistical differences between the populations in different centers regarding the variables that were clinically associated with the presence of anemia. Therefore, in our original manuscript, we constructed a multivariate model that used those variables as the determinants of anemia (page 9, paragraph 4, line 1-7) (Table 3). After appropriate statistical correction, ‘center’ was defined as an independent risk factor for the presence of anemia. Following the comments of the reviewer, we included additional information to clarify this issue in the revised manuscript (page 4, paragraph 1, line 8-17).

The following are minor comments, based on STROBE checklist:

9) BIAS, in the methods section, potential sources of bias are not addressed

Response: Following the suggestion of the reviewer, we added bias part in the original manuscript (page 3, paragraph 3, line 1-5). 

10) Explain how you arrived at this study size

Response: We did not perform a formal sample size calculation. We intended to recruit all consecutive eligible patients for a duration of seven months. 

12.a) You should be more precise to describe the logistic regression: you decide to exclude patients with malignacy and PHD form the analysis. this should be stated in the methods section and the choice must be justified. the criteria used to select the variable for the multivariate analysis must be specified in the methods.

Response: We wanted to examine the factors associated with anemia of CKD and we excluded the patients with malignancy and primary hematological disease, since both conditions were known to be associated with anemia (reference no: 14). Following the comment of the reviewer, we moved the relevant information to the Methods section in the revised manuscript (page 4, paragraph 1, line 8-17). 

12.c) How do you handle missing data?

Response: We didn’t fill-in or impute the missing values in the statistical methods, but rather omitted it. The frequency of missing data was <10% (ferritin 5%; iron 8%; CRP 2%) in the clinical parameters regarding anemia. 

16) the presentation of the results of the binary logistic regression is not clear: OR (and not just p value should be reported in the text). You specified a reference level of ferritin < 100, but no other reference level for other continuous variable such as CRP e GFR and Iron are present.

Response: According to the current guidelines, a ferritin level of above 100 ng/mL reflects the presence of appropriate iron stores (National Kidney Foundation. K/DOQI clinical practice guidelines for anemia of chronic kidney disease. 2000. Am J Kidney Dis 2001;37(Suppl 1):182-238). Therefore, we used a cut-off value for ferritin. However, to the best of our knowledge, there is no definite cut-off value for CRP and eGFR that indicates renal anemia. Following the comments of the reviewer, OR data was added to the manuscript. 

18) your results shows that ESA is underused, this is interesting and should be cited in the discussion. Moreover, is there any significant difference among centers on this point?

Response: We appreciate this comment. In line with this suggestion, we discussed about underused ESA treatment and the possible causes (page 11, paragraph 3, line 1-5).

20) the authors interpretate the impact of treatment center on anemia as due to the socio-economical differences in the population. it sounds strange if we think that the three center are located in the same area. Please clarify.

Response: The three centers are located in the same geographical region, namely Marmara. However, one of them is located in a different city called Kocaeli. The remaining two centers are located in different districts of Istanbul which is a very large city with a population of nearly 16 million inhabitants. There are well defined socioeconomical differences between different districts of Istanbul. 

REVIEWER #4

The authors have conducted an observational study to study prevalence of and management of anemia in non-dialysis dependent CKD patients. First the title of the study is very vague. Methodology section is very weak, there is no explanation of study design, study instrument, recruitment of patients, inclusion exclusion criteria, methodological flowchart etc. Statistical methods used in current study are very basic and gives very basic information. Authors could have performed more analysis especially for comparison of management in different centers with respect to patient outcomes. There are hundreds of articles on this topic online, i feel this authors literature review is bit weak. I would suggest them to read relevant articles and modify their paper accordingly.

Response: In our revised manuscript, we made changes according to the specific comments of the academic editor and other reviewers. We added additional information to reinforce methodology and clarify study design (page 3, paragraph 1, line 2-7; page 3, paragraph 3, line 1-5; page 4, paragraph 1, line 8-17). We used the appropriate statistical methods to analyze our data including multivariate logistic regression and multigroup comparison with ANOVA test, which are well beyond the basic level. If there is any further specific statistical method that the reviewer suggests, we will be happy to incorporate them to our study in order to make a more sound analysis. Following the comments of the reviewer, we added additional references to reinforce our literature review (reference no: 9,10,11,14,24) and made appropriate modifications.

---

## [Editor Report · Decision Letter 1]

13 Mar 2020

Prevalence of anemia in predialysis chronic kidney disease: is the study center a significant factor?

PONE-D-19-32342R1

Dear Dr. Seyahi,

We are pleased to inform you that your manuscript has been judged scientifically suitable for publication and will be formally accepted for publication once it complies with all outstanding technical requirements.

With kind regards,

Martina Crivellari

Academic Editor

PLOS ONE
---

## [Editor Report · Acceptance letter]

18 Mar 2020

PONE-D-19-32342R1 

Prevalence of anemia in predialysis chronic kidney disease: is the study center a significant factor? 

Dear Dr. Seyahi:

I am pleased to inform you that your manuscript has been deemed suitable for publication in PLOS ONE. Congratulations! Your manuscript is now with our production department. 

With kind regards,

on behalf of

Dr. Martina Crivellari 

Academic Editor

PLOS ONE